## [Peer Review File · Nature Communications]

Optics miniaturization strategy for demanding Raman spectroscopy applicationsREVIEWER COMMENTS

Reviewer #1 (Remarks to the Author):

The manuscript reports on a new small device developed for various applications based on Raman spectroscopy. Limitations of commonly used Raman instrumentations and existing miniaturized systems are extensively described in the introductory part, followed by a detailed description of the design and working principle of the developed miniaturized Raman spectrometer. Finally, the successful use of the new miniature Raman system either handheld or coupled to another device (microscope) for qualitative and quantitative purposes is demonstrated.

Although a detailed description of the construction of the new Raman system is given, information about the software used is missing. Especially if one considers the fact that in order to overcome spectral shortcomings caused by the miniaturization of the instrumental setup, several preprocessing techniques were developed that need to be applied. Did the authors develop the new software themselves or did they implement the new algorithms in a commercial one? Are spectral processing procedures time-consuming? What are the costs of the construction of the device? How is the device powered? Given the spectral resolution of 7 cm⁻¹, comparable to portable Raman spectrometers, is the new device suitable for fundamental research or is it more applicable for routine measurements?

I recommend the manuscript for publishing in Nature Communications after addressing the above issues.

Reviewer #2 (Remarks to the Author):

The authors of the manuscript „Optics miniaturization strategy for demanding Raman spectroscopy applications” describe the structure, functionality and various test results of a miniaturized Raman spectrometer, which was used to detect low concentrations of methanol, drugs and bacteria. The demonstrated levels of detection (LoD) are competitive with other methods like HPLC or classic bulk Raman-spectroscopy. At the same time the size of the overall spectrometer is strongly reduced to a few centimeters. This is indeed impressive and of practical importance as one can envisage that such spectrometers can be implemented in standardized monitoring systems for e.g. the food industry. On the other hand, the spectrometer represents rather a well designed and optimized combination of already known and tested elements and corrections methods. There is not really a new phenomenon or measurement method demonstrated.

In my opinion the manuscript is therefore not suited for Nature communications but should be published in a more specialized journal. I suggest to refer it to Nature methods or Nature photonics or another journal specialising in reporting on gradual advancement of instruments and measurement methods.

In addition I suggest the authors clarify the following points:

1. The authors give a very nice overview of the current state of the art of Raman measurement system. I suggest to move the general text about the requirements for a miniaturized Raman-system (lines 125-133) towards the front of the main section so that the detailed description of the pros and

cons of the different elements can be discussed with respect to these purposes.

2. Lines 104 to 124 give an overview of already realized miniaturized Raman spectrometers, which use the same or similar measurements principle as the one of the authors. The system of the authors should therefore suffer in principle from the same problems as outlined here. The authors should state therefore clearly in which respect and how their system is an advance and which limits are surpassed to make a case for their prototype.

3. What is the size of the new Raman-measurement system? It can be seen in Fig.S28 but the authors should also give its dimensions in cm or mm, so that reader can compare it directly with the dimensions give in lines 104- 124. Just to say that it is miniaturized to “several cm” in size (line 153, 154) is not enough.

Later a size of “10x 15x3mm” is given (line 525). Is this the size of the prototype used for the test measurements or is this a future design which is not yet realized?

4.section “Design and working principle”: There should be some more data given about the Bragg grating. Is it a volume Bragg grating? How many lines/layers? What is the distance from the grating to the sensor, so that the reader can recognize that indeed the claimed resolution can be achieved with the give pixel size of the CMOS sensor.

5. section “Design and working principle”: Is there a reflected laser beam (ghost beam) from the reference towards the sample (offset a bit from B3)? If yes, does that cause problems (additional signal on the sensor)?

6. The sentence “our basic strategy ...” (lines 492-494) should be moved at the beginning of the discussion section to act as a general starting point from where details are derived.

7. Supplements: state the methanol concentrations in Fig. S13 and Fig. S15 for the different curves

Optics miniaturization strategy for demanding Raman spectroscopy applications

Oleksii Ilchenko^{1,2}, Yurii Pilhun^{2,3}, Andrii Kutsyk^{2,3,4}, Denys Slobodianiuk^{3,5}, Yaman Goksel¹, Elodie Dumont¹, Lukas Vaut¹, Chiara Mazzoni¹, Lidia Morelli¹, Sofus Boisen², Konstantinos Stergiou², Yaroslav Aulin², Tomas Rindzevicius¹, Thomas Emil Andersen⁶, Mikael Lassen⁷, Hemanshu Mundhada⁸, Christian Bille Jendresen⁸, Peter Alshede Philipsen⁹, Merete Hædersdal⁹, Anja Boisen¹

¹Technical University of Denmark, Department of Health Technology, Center for Intelligent Drug Delivery and Sensing Using Microcontainers and Nanomechanics, Kgs. Lyngby 2800, Denmark

²Lightnovo ApS, Birkerød 3460, Denmark

³Taras Shevchenko National University of Kyiv, Kyiv 01601, Ukraine

⁴Technical University of Denmark, Department of Energy Conversion and Storage, Kgs. Lyngby 2800, Denmark

⁵Institute of Magnetism, Kyiv 03142, Ukraine

⁶ Department of Clinical Microbiology, Odense University Hospital and Research Unit for Clinical Microbiology, University of Southern Denmark, Odense, Denmark

⁷ Danish Fundamental Metrology, Hørsholm 2970, Denmark

⁸Cysbio ApS, Hørsholm 2970, Denmark

⁹Department of Dermatology, Copenhagen University Hospital, Copenhagen 2400, Denmark

Abstract

Raman spectroscopy provides non-destructive, label-free quantitative studies of chemical compositions at the microscale as used on NASA's Perseverance rover on Mars. Such capabilities come at the cost of extremely high requirements for instrumentation.

Here we present a centimeter-scale miniaturization using cheap non-stabilized laser diodes, densely packed optics, and non-cooled small sensors, while the performance is comparable with expensive bulky research-grade Raman systems. It has excellent sensitivity, low power consumption, perfect wavenumber, intensity calibration, and 7cm^{-1} resolution within the $400\text{-}4000\text{ cm}^{-1}$ range using a built-in reference. We present solutions to Raman miniaturization challenges: laser temperature and power stabilization, reduction of sensor dark noise, compensation on pixel-to-pixel quantum efficiency variation, laser optical isolation and high spectral resolution. Moreover, shifted-excitation Raman difference spectroscopy (SERDS) and spatially offset Raman spectroscopy (SORS) functions are compatible.

High performance and versatility are demonstrated in use cases including quantification of methanol in beverages, *in-vivo* Raman measurements of human skin, quantification of *p*-coumaric acid and serine in bacterial fermentations, chemical Raman mapping at sub-micrometer resolution, quantitative SERS mapping of the anti-cancer drug methotrexate and *in-vitro* bacteria identification by Raman mapping. We foresee that the miniaturization will allow realization of super-compact Raman spectrometers for integration in smartphones and medical devices, democratizing Raman technology.

Main

The discovery of the Raman effect in 1928 has not only aided to the fundamental understanding of the quantum nature of light and matter interaction but has also opened completely novel areas of optics and spectroscopic research that has accelerated greatly during the last decade. The utility of Raman spectroscopy has been demonstrated for a diverse and wide range of biological, biomedical and chemical applications, such as chemical imaging of living cells and tissues¹, stem cell² and cancer research³, bacterial identification⁴⁻⁷, chemical hazards and illicit substances detection⁸, as well as food and product authentication⁹, and with a great deal of interest and research into its potential for disease diagnosis in the laboratory¹⁰⁻¹² and *in-vivo*¹³⁻¹⁶. Raman

spectroscopy has been developed into a variety of methods and experimental realizations, such as confocal Raman microscopy, Raman endoscopy, spatially offset Raman spectroscopy (SORS), resonance Raman spectroscopy, and surface enhanced Raman spectroscopy (SERS). The listed Raman spectroscopy and microscopy methods are non-destructive, label-free, non-invasive, and capable of providing 3D molecular sensing with depth profiling. Such capabilities, however, come at the cost of extremely high requirements for instrumentation: i) the used laser should have stable wavelength and stable high optical power, ii) the spectroscopic sensor should have low noise, and iii) the spectrometer's optics should have a large clear aperture. Therefore, Raman spectroscopy and microscopy applications that require high spectral resolution and sensitivity would normally need to be performed on high-end, bulky, and costly Raman instruments. The need for miniaturization of Raman instrumentation is driven by applications where the complexity and/or the bulkiness of existing devices is obstructive. Application examples in need of miniaturization include space exploration¹⁷⁻¹⁹, on-site toxic substance inspection²⁰⁻²², *in-vivo* diagnostics of tissues^{23,24}, chemical identification in hardly accessible places using robots²⁵ and drones, and Raman device integration into robotic arms for biomedical applications²⁶⁻²⁸.

Multiple optics miniaturization strategies for Raman spectrometers have been proposed within the last decade. More conventional solutions are based on bulky dispersive optics, that require the presence of a slit or a pinhole and a grating or a prism as a dispersion element. Here, the miniaturization comes at the cost of e.g. reduced spectral range and resolution, reduced confocality (mostly relevant for microscopy applications) and sensor sensitivity²⁹. On-chip spectrometers have the potential to offer dramatic size, weight, and power reductions compared to bulky optics-based instruments³⁰. Multiple extra-compact on-chip spectrometer concepts have been proposed³¹. For example, on-chip dispersive spectrometers based on Echelle gratings³², planar concave gratings³³ and arrayed waveguide gratings³⁴⁻³⁷ provide low-cost solutions for on-chip spectral analysis using visible and near-infrared light. However, when applied to high-resolution spectrum acquisition, these devices suffer serious signal-to-noise ratio (SNR) penalties as a result of spreading the input light over many spectral channels. Furthermore, the device footprint and system-level complexity increase linearly with the number of spectral channels, since the spectral resolution scales inversely with the optical path length (OPL) and each channel requires a dedicated photodetector.

Unlike dispersive spectrometers, Fourier transform (FT) spectrometers overcome the trade-off between SNR and spectral resolution benefiting from the multiplex advantage, also known as Fellgett's advantage³⁸. Traditional benchtop FT spectrometers use moving mirrors to generate a tunable OPL, a design not readily amenable to planar photonic integration. On-chip FT spectrometers instead rely on thermo-optic or electro-optic modulation to change the OPL in a waveguide. However, this concept suffers from poor scaling because the number of spectral channels is equal to the number of switches and photodetectors.

On the other hand, on-chip digital FT spectrometry could support high-resolution spectra and high SNR via time-domain modulation of a reconfigurable Mach-Zehnder interferometer³⁹. This approach requires only a single-element photodetector rather than a linear detector array, which reduces cost and system complexity. However, digital FT spectrometry has only been demonstrated in a very narrow spectral range³⁹ that limits its applicability in Raman spectroscopy applications.

A promising on-chip Raman spectrometry concept is CMOS based hyperspectral (HS) filter technology. The HS filter technology is already used in various commercially available HS cameras for imaging applications in the VIS-NIR (470-900nm), NIR (600-970nm) and SWIR range (1100-1700nm) to study the reflectance or transmission spectra of imaged targets. However, existing proof of concept experiments in the field of Raman spectroscopy demonstrate low spectral resolution (2.4 and 3.0nm) which significantly limits the application areas⁴⁰. Moreover, HS filter technology has significantly lower Raman signal throughput in comparison with dispersion-based approaches due to reduced transmittance of Fabry-Pérot narrow-band optical filters applied on top of the sensor pixels. Another limitation comes from the need to illuminate a much larger sample area while maintaining equal laser power density to reach a sensitivity comparable to dispersion-based spectrometers. This happens since each pixel on the HS sensor collects signals from a different area on the sample. In particular, it could lead to sample overheating and/or significantly increased fluorescence level for Raman spectroscopy-based studies on fluorescent materials similar to wide-field Raman microscopy^{41,42}.

Another promising concept is based on spatial heterodyne Raman spectrometry (SHRS)^{43,44}. This technology has not yet been realized on-chip, nevertheless, existing prototypes demonstrate research grade performance in terms of spectral range (50-3300cm⁻¹), resolution (5-9cm⁻¹) and SNR⁴⁴. Here, high resolution can

be realized without using long focal length dispersing optics. Therefore, it is possible to use monolithic construction techniques to make a compact and robust device. SHRS could be an ideal candidate for an optics miniaturization strategy in many aspects, except for issues related to increased level of stray light and non-confocality in comparison with slit/pinhole-based dispersion spectrometers.

Due to the mentioned drawbacks of emerging technologies, challenging Raman spectroscopy and microscopy applications are still relying on conventional bulky dispersion optics. Here, we can select between miniaturization concepts based on reflective⁴⁵⁻⁴⁷ and transmission gratings^{29,48}. Designs based on reflective gratings could be as compact as $17 \times 15 \times 8 \text{ mm}^3$ ⁴⁹, however they suffer from low spectral resolution ($>90\text{cm}^{-1}$)⁴⁹. A relevant compromise could be found by using a spectrometer footprint of around $40 \times 42 \times 24 \text{ mm}^3$ that provides spectral resolution of around 25cm^{-1} within a spectral range of $200\text{-}2000\text{cm}^{-1}$ ⁴⁹. Raman spectrometer designs based on transmission Bragg gratings benefit from higher throughput (up to 97% of diffraction efficiency in broad NIR range), alignment robustness and minimized stray light in comparison with reflective gratings⁵⁰. On the other hand, existing reflective and transmittance grating designs use a line or imaging sensor with a typical pixel size between $10\text{-}20\mu\text{m}$. Such sensors should have 1000-2000 pixels in dispersion domain in order to cover a relevant Raman shift range between $400\text{ to }2000 \text{ cm}^{-1}$. This leads to a typical sensor length of 12 to 28mm ^{29,48} which can be another limitation for spectrometer miniaturization. It is important to mention that larger pixel size (typically more than $12\mu\text{m}$) of CCD and CMOS sensors results in increased dark noise collected by the sensor, therefore, sensor cooling down to at least $+5^\circ\text{C}$ is recommended in order to reach high sensitivity during collection of weak Raman signals^{51,52}. Another important problem that limits the sensitivity of Raman measurements is caused by variation of quantum efficiency (QE) of the individual pixels, which usually is at the level of 0.1-0.5%^{53,54}. Unfortunately, pixel-to-pixel QE variation is not static in time and cannot be compensated via postprocessing for a long period of time⁵³. This problem can be successfully solved via full vertical binning over the pixels of the imaging sensor⁵⁵, however, it would require deep cooling of the imaging sensor to avoid an increase of dark noise summed up from many pixels in vertical arrays. The here listed problems significantly limit miniaturization possibilities if a high Raman spectrometer performance should be maintained.

Existing Raman miniaturization strategies, suffer from one or more of the following issues: insufficient spectral resolution and/or spectral range, limited SNR, due to high level of sensor dark noise, pixel-to-pixel QE variation on sensor, poor confocality or depth sectioning, instability of laser wavelength and laser optical power, high laser optical feedback sensitivity, and high-power consumption. Here, we present a miniaturization strategy that allow us to solve the listed problems and create extra-compact Raman spectrometers and microscopes based on non-stabilized laser diodes, close-packed optics, and non-cooled small pixel size sensors. The achieved performance is comparable with research-grade Raman systems. Our proposed miniaturization concept is based on real-time calibration of Raman shift and Raman intensity using an in-built reference channel that collects the Raman spectrum of polystyrene located in the spectrometer. Implementation of a reference channel has been realized in UV-VIS absorption spectroscopy decades ago⁵⁶. However, we are not aware of any successful realization of a reference channel in Raman spectroscopy. Existing approaches are usually based on placing a reference sample in the place where the laser beam delivery optical path is overlapped with the back scattered Raman signal. Typically, the reference material is located just before the sample probe (for example, a flat window or lens sub-assembly made of CaF_2)⁵⁷. Such a realization makes it possible to calibrate the laser wavelength and intensity using a Raman peak of the reference sample. However, this method fails if the Raman spectrum of the measured material contains Raman peaks or fluorescence signal within the same spectral range as the reference sample. At such conditions real-time laser wavelength and intensity calibration is problematic - if at all possible. In our concept, the reference channel is independent of the main optical path, which eliminates any interference between reference and main Raman signal collection channels.

Here, we demonstrate miniaturization down to several centimeters (optical part dimensions: $7 \times 2 \times 0.8\text{cm}$), an achieved limit of detection (LoD) down to 0.07% of methanol in water-ethanol solution, a low power consumption of around 2 Watts, perfect wavenumber ($\pm 1.5\text{cm}^{-1}$) and intensity calibration ($\pm 1\%$) combined with high spectral resolution of around 7 cm^{-1} within the broad spectral range of $400\text{-}4000\text{cm}^{-1}$. The high performance and vast versatility offered by our strategy facilitate simple integration into various instruments and applications. As use case examples, we show applications within quality control of alcoholic beverages, quantification of nutrients and metabolites during bacterial fermentation, *in-vivo* measurements of human skin, therapeutic drug monitoring and *in-vitro* bacteria identification.

Results

Miniaturization of Raman systems includes size-reduction of (i) spectrometer, (ii) Raman beam delivery path, (iii) laser beam delivery path, (iv) beam splitting unit and (v) sampling optics. Thus, besides the spectrometer, miniaturization also impacts the choice of lasers. In particular, proper operation of diode lasers requires accurate temperature stabilization of the active element⁵⁸ and optical power stabilization⁵⁹. Compromises on stabilization of these parameters lead to laser frequency drift, mode-hopping and power instability⁵⁸⁻⁶². Additionally, diode lasers are also sensitive to the external optical feedback or back reflections of laser irradiation from laser beam delivery elements and sample⁶³⁻⁶⁷. This effect leads to increased laser power and wavelength instability and could even permanently damage the laser diode⁶⁸. Therefore, the laser diode requires optical isolation typically realized with Faraday rotator⁶⁹. This is an expensive element which limits miniaturization capabilities. We begin the description of our proposed miniaturized Raman spectrometer with the laser related problems.

Design and working principle. Optical scheme of the developed miniaturized Raman spectrometer is shown in **Figure 1a**. Here, a typical AlGaAs laser diode in a Ø5.6 mm TO package with Fabry–Pérot resonator at a central wavelength of 785nm and a maximum power of 200mW is used as a Raman source (L1). Laser spectral linewidth at half maximum (LWHM) is 0.2nm which is sufficient to obtain a spectral pixel resolution of the miniaturized Raman system of 0.3nm. The selected type of diode laser usually requires precise temperature stabilization for Raman spectroscopy applications to prevent laser wavelength drift and “mode hop”. In order to avoid bulky, costly and high-power Peltier elements for temperature stabilization of the laser diode, we propose a concept that does not require laser wavelength stabilization at all. According to **Figure 1a**, a collimated laser beam is split into two beams using prism (P1); first part of the split beam B1 is focused on a polystyrene sample that is glued to a special Raman edge filter F3 coated with an aluminum mask (the mask serves as a spectral slit), second part of the split beam B2 is focused on the slit and reflected from the Raman filter F3 towards the sample as beam B3 (**Figure 1b**). As a result, two Raman spectra (main channel and reference channel) in the “fingerprint” range (400-2700cm⁻¹) are simultaneously collected by NIR enhanced imaging CMOS sensor in the range 800-960nm (**Figure 1c**).

The Raman beam delivery system consists of Raman probe L5, slit lens L4 (f = 30mm) and spectrometer (elements F3, L6 (f=30mm), F4, Grating, L7 (f=6.2mm), Sensor). The total distance between grating and sensor is 8.5mm. The spectral slit size is 25µm that zooms down to 5.4µm on CMOS sensor focal plane with binned pixel size of 4 µm. Imaging capabilities of L7 provide uniform resolution along the spectral dimension on CMOS sensor at diffraction limited spot size. This makes it possible to concentrate most of the Raman signal intensity into a single row on the CMOS sensor (**Figure 1c**). Spectrometer is equipped with fused silica transmission Bragg grating with 1500 lines/mm and average efficiency in the first order of diffraction ~96% in the range 800-960nm⁷⁰. In combination with NIR coating for all optical elements, entire optical system has extremely high throughput from the sample to the detector ~92%. Listed features significantly boost sensitivity of miniaturized Raman spectrometer. The Raman beam delivery system is optimized in terms of minimization of the stray light that could appear from the laser beam reflected/scattered from the reference sample and cause artifacts in the data channel. This is realized via implementation of corrugated surfaces and back anodizing on the spectrometer housing.

In order to cover a “high frequency” Raman range, we have added an extra AlGaInP laser diode with Fabry–Perot resonator L2 with a central wavelength of 675nm, LWHM 0.2nm and a maximum power of 200mW. This additional laser L2 and the main laser L1 are switched on sequentially, providing two different Raman shift ranges with the same grating. The proposed approach makes it possible to collect in the “high frequency” Raman range by the same optical elements in the same spectral range 800-960nm that is used for collection of the “fingerprint” range. This strategy allows us to maintain a high SNR for Raman spectra in “high frequency” range due to relatively high QE of the CMOS (60% at 840nm, 40% at 940nm) sensor in the range 800-960nm⁷¹. The collimated beam from non-temperature-stabilized laser L2 is combined and coaligned with collimated beam from L1 by dichroic mirror D1. After D1, laser irradiation from L2 propagates through the same optical path as B1-B3 and targets the polystyrene sample on the slit S_{ref} and sample of interest S_{data} . Finally, two Raman spectra of the main channel and the reference channel in the range of 2700-4000cm⁻¹ are simultaneously collected by the imaging CMOS sensor (**Figure 1d**). Therefore, the miniaturized Raman spectrometer can collect a combined

Raman spectrum in the range of 400-4000 cm^{-1} reaching the performance typically associated with much larger, research grade systems.

Electronics and software. Initial processing of raw data obtained from the spectroscopic camera and converting it into calibrated Raman spectrum is implemented in an embedded computer of the miniaturized Raman device. The device is controlled by a microcomputer running ARM processor with clock frequency 1GHz and 512MB RAM having CSI interface for attaching CMOS camera, and several interfaces for accessing the device from PC or smartphone/tablet, such as USB, Bluetooth and Wi-Fi. The embedded microcomputer is also controlling lasers and provides indications on the device itself. The device is powered by 1000 mA·h Li-ion battery and consumes about 200 mA in idle mode and up to 600 mA with laser on. This provides an active operation time of up to 2 hours (depending on laser current set) and up to 5 hours in idle mode. All spectrum pre-processing such as binning, hot pixels removal, real-time wavelength calibration, intensity normalization, and laser mode hopping correction is done on the embedded computer inside the device with custom algorithms written in C++ language. The embedded processor is powerful enough to perform required processing in real-time on each spectrum acquisition. Only a few rows of CMOS sensor image are processed to extract spectra, therefore, computational requirements are not high, which allows to keep power consumption low. In the present device the microcomputer can process up to 50 frames per second, much faster than actually needed, considering that exposure times for Raman collection are usually longer than 0.1 s. Electronics occupy about 95×31×15 mm in the device, and consists of two PCBs: commercially available ARM single-board computer and custom-made laser driver and battery controller board. The size of electronics can be significantly reduced (at least three times) by developing a fully customized solution. Processing algorithms also could be optimized and embedded into FPGA or ASIC instead of ARM processor to reduce power consumption even more.

Subsequent data processing and analysis, such as identification and quantification, are performed on PC using custom acquisition software for data collection and MATLAB application for advanced processing (Matlab R2020b (license provided by Technical University of Denmark)).

Wavenumber and intensity calibration. Raw Raman spectra from the reference and main channel collected as a function of time are shown in **Figure 1e-i**. Time-lapse experiment clearly shows peak shift of polystyrene (reference channel) and polypropylene (main channel) caused by laser wavelength drift. Since both reference and main channel were collected from the same laser source and acquired simultaneously by the imaging CMOS sensor, we could apply wavenumber calibration for each collected Raman spectrum in the main channel. Calibration is done by peak fitting of several prominent polystyrene peaks in the reference channel and correcting for their known position, with the same correction applied also to the main channel. The result for wavenumber calibration versus time is shown in **Figure 1k**, demonstrating high and stable calibration accuracy ($\pm 2\text{cm}^{-1}$) using multiple peaks of polystyrene measured by the main channel. At the same time, **Figure 1l** demonstrates that Raman intensities in the reference and main channels are correlated as well (black and red curves). Therefore, it is possible to normalize main channel for laser power fluctuations during each spectrum acquisition (blue curve in **Figure 1l**). This feature provides an excellent solution for quantitative Raman spectroscopy applications where laser intensity monitoring is required to reduce concentration determination error⁷².

“Mode hop” deconvolution. The described wavenumber and laser intensity calibrations fail if a fast laser wavelength change occurs due to laser “mode hop.” In this case, even at short exposure times (<0.2sec), spectra will be collected at reduced spectral resolution and decreased peak intensity because the energy is spread over several individual pixels. Nevertheless, this problem can be solved by spectral deconvolution of the main channel based on the known spectral profile of the Raman spectrum of polystyrene in the reference channel (**section S1 in SI, Figure S1-S6**). **Figure S5** demonstrates that the deconvolution procedure significantly helps in recovering the original spectral resolution and Raman intensity during a “mode hop” process.

Spectral deblurring and spectral resolution. Pixel limited spectral resolution of miniaturized Raman spectrometer is 5.6-3.4 cm^{-1} in the range of 400-2000 cm^{-1} (from laser excitation at 785nm) which corresponds to the optimal slit size $\sim 18\mu\text{m}$. Nevertheless, we decided to increase the actual slit size up to $25\mu\text{m}$ for increased signal throughput, which transfers into the calculated spectral resolution of 10.2-5.8 cm^{-1} . Deviation from diffraction limited optics design would lead to even worse resolution. However, it is possible to measure the spectral apparatus function of the entire Raman system and compensate on it using deblurring methods⁷³. Raman spectrum of diamond with natural linewidths of around 1.8 cm^{-1} was used for the determination of the spectral apparatus function⁷⁴ (**Figure S7-S9**). The result of spectral deblurring is shown in **Figure S10-S12, section**

S2 in SI. Figure 1m illustrates improvements in terms of spectral resolution after correction on “mode hop” and spectral deblurring (black curve: before corrections, red curve: after corrections) demonstrating final high spectral resolution $\sim 7\text{cm}^{-1}$ in worst case scenario (Figure S10). Spectral resolution of miniaturized Raman spectrometer was tested according to the procedure described in the ASTM E2529 – 06(2014) standard. The spectrum of calcite reference sample was acquired (**Figure S13**). The shape of 1085 cm^{-1} calcite line was fitted with a Voigt function (**Figure S14**) with 7.4 cm^{-1} full width at half maximum. The spectral resolution was further estimated by the formula from ASTM standard as 6.6 cm^{-1} .

Shifted-excitation Raman difference spectroscopy (SERDS). Laser wavelength drift, that is compensated with the use of the reference channel, but still is present due to laser instability, can be used for fluorescence subtraction similar to SERDS⁷⁵. The method is based on the fundamental difference between Raman and fluorescence properties with regards to the wavelength shift of the excitation source. It is based on the fact that the fluorescence spectrum position is independent from the shift of the excitation wavelength, whereas the Raman spectrum is shifting along the shift of the excitation wavelength. This enables us to distinguish between Raman and fluorescence spectra. Typically, a dual- or tunable laser source is required for SERDS⁷⁶. In our device, a non-stabilized laser automatically provides a gradual wavelength shift due to the laser body heating during operation. **Figure 1n, 1p** show shifted-excitation Raman spectra of polypropylene at different time points from the experiment in **Figure 1f, 1i**. Here, we applied SERSD algorithm⁷⁶ for fluorescence profile restoration. **Figure 1o, 1q** demonstrate fluorescence free Raman spectra (red curves) obtained after subtraction of resolved fluorescence profiles (green curves) from initial Raman spectra (black curves). Hereby we demonstrate that a non-wavelength-stabilized laser source can also be used for efficient fluorescence subtraction via SERDS if the device has a reference channel to compensate a wavelength shift in the Raman spectrum.

These multiple pre-processing techniques have been developed in order to realize a miniaturized Raman spectrometer, relying on the presence of a built-in calibration channel. The resulting device provides a (i) fluorescence-free, (ii) “mode hop”-free, (iii) laser power fluctuations-free Raman spectrum in the range $400\text{-}4000\text{cm}^{-1}$ with a spectral resolution $\sim 7\text{cm}^{-1}$ (**Figure 1g, 1j**). Device control and data transfer can be performed by wire or wirelessly with software installed on smartphone/tablet or PC.

Figure 1. Optical layout and working principle of miniaturized Raman system. **a)** Optical scheme of miniaturized Raman system, **b)** reference sample on filter/slit, **c), d)** CMOS sensor image that demonstrates simultaneous acquisition of main and reference Raman signals from laser excitation at 785nm (c) and 675nm (d); CMOS images represent Raman spectrum of water-ethanol solution (60:40) in the main channel and Raman spectrum of polystyrene in the reference channel, **e), f), h), i), j)** laser stability experiment versus time that represents raw Raman spectra variation of polystyrene in the reference channel under laser excitation wavelength 785nm (e) and 675nm (h) and variation of raw Raman spectra of polypropylene in the main channel under laser excitation wavelength 785nm (f) and 675nm (i), **g), j)** Raman spectra of polypropylene in the main channel versus time under laser excitation wavelength 785nm (g) and 675nm (j) after real-time (i) Raman shift, (ii) Raman intensity calibrations, (iii) anti “mode hop” deconvolution and (iv) spectrum deblurring being applied, **k), l)** time dependent experiment of polystyrene sample placed in the main channel that demonstrates long time calibration stability of the Raman shift (k) and Raman intensity (l), **m)** Raman spectrum of toluene before (black curve) and after (red curve) multiple preprocessing procedures being applied, **n), p)** three Raman spectra of polypropylene at different time points under a laser excitation wavelength of 785nm (n) and 675nm (p), **o), q)** Raman spectra of polypropylene before (black curve) and after SERDS correction (red curve) under laser excitation wavelength of 785nm (o) and 675nm (q).

Sensor dark noise. The sensitivity of the Raman spectrometer strongly depends on the dark noise of the detector. Due to the weakness of the Raman signal, most of spectrometers are equipped with cooled linear or imaging sensors with relatively large pixel size (12-25 μm). Sensor cooling reduces the dark noise whereas large pixel size allows one to collect more photons maintaining high resolution at the same time⁷⁷. Nevertheless, this is a high power demanding and bulky approach. As a solution towards miniaturization without significant compromise on sensitivity, we implemented a CMOS sensor with a small binned pixel size of 4 μm and managed to compress the signal from the Raman spectrum into a single row on the sensor using high numerical aperture (NA) imaging lens L6 (**Figure 2a**). Signal compression allows us (i) to maximize SNR per pixel and (ii) avoid averaging of additional rows with unwanted additional dark noise. This is illustrated in an experiment where equal amount of total intensity of SERS signal was distributed over 20 rows on the CMOS sensor (**Figure 2b**). Comparison of SERS spectra of trans-1,2-bis(4-pyridyl)ethane (BPE) in **Figure 2c, 2d** highlights 3 times higher SNR when the SERS signal is compressed into a single row. Signal to noise ratio was estimated from the spectrum of polystyrene reference sample at laser power 100mW, exposure time 0.25s, number of repetitions 10. The amplitude of 1001 cm^{-1} was used as signal level, while the noise level was estimated as root mean square of the noise in the spectral range 1700-2100 cm^{-1} from the same spectrum. The signal to noise ratio obtained in such a way is 1256 (**Figure S15-S16**).

Pixel QE variation. The fact that miniaturized Raman spectrometer uses non-wavelength-stabilized laser allows us to compensate on another sensitivity-limiting factor which is pixel-to-pixel QE variation of the spectroscopic sensor (**Figure 2e**)^{53,54}. **Figure 2f** represents a fluorescence spectrum from a glass cover slide excited by a laser with an excitation wavelength of 785nm obtained after averaging of 10 repetitions. It is visible that the spectral profile of fluorescence contains noise-like spikes. This “noise” is always present no matter how long a spectrum is collected or how many repetitions are applied because it represents pixel-to-pixel QE variation. However, once the reference channel-based wavenumber calibration is applied, pixel-to-pixel QE variation is significantly reduced (**Figure 2g**). It happens because each spectrum wavenumber corresponds to a different pixel in the sensor row when the laser wavelength is shifted. As a result, pixel-to-pixel QE variation is averaged out over the pixels in the same row.

Laser optical isolation and SORS. Typically, laser diodes with Fabry-Pérot resonator require expensive optical isolation based on the Faraday effect⁶⁹. A less complex solution may be based on laser polarization rotation via quarter waveplate; however, it provides reduced attenuation of the back reflected signal (<20dB)⁷⁸. Here we implement an off-axis laser beam delivery approach that (i) avoids laser back reflections targeting the laser aperture and (ii) boosts miniaturization capabilities (**Figure 2h**). In addition to laser optical isolation, off-axis laser beam delivery supports Spatially offset Raman spectroscopy (SORS) conditions that allow us to avoid unwanted fluorescence contribution from out-of-focus layers inside the sample⁷⁹. Effectively, it spatially separates contribution from out-of-focus regions and in-focus signal on the imaging sensor. **Figure 2i** shows our miniaturized Raman spectrometer measuring whiskey content through the glass bottle. In order to demonstrate SORS performance we performed two experiments: with the laser beam aligned on-axis (black curve) and off-axis (red curve) (**Figure 2j**). Black spectrum clearly demonstrates SORS benefit to avoid the contribution of fluorescence from the glass bottle (see **movie 1**).

Transmittance correction. The transmittance correction was applied by acquiring the spectrum of a black body standard source (tungsten halogen lamp) at 3000K (**Figure S17**).

Application I: Quantification of toxic methanol in vodka. To demonstrate the sensitivity and quantification performance of our miniaturized Raman spectrometer we performed measurements of vodka samples with different concentrations of methanol (see **Figure S18-S21, Table S1-S2**). The raw Raman spectra of water-ethanol solutions with different concentrations of methanol (variation between 0-40%) in the range 400-2300 cm^{-1} and 2750-4000 cm^{-1} are shown respectively in **Figure 2k, 2n**. Results of PLS calibration for methanol quantification demonstrate LoD = 0.07%, and LoQ = 0.25% (**Figure 2l-2p**). To the best of our knowledge, the lowest previously reported LoD that was obtained by research grade Raman spectrometer with deep cooling CCD was 0.23–0.39%⁸⁰. However, according to European regulations, methanol concentration in vodka products should be below 0.5%⁸¹. This means that LoD should be below 0.1% to perform routine methanol quantification through the bottle with vodka. Now, such methanol control becomes possible with our highly sensitive miniaturized Raman spectrometer.

Figure 2. Demonstration of sensitivity and quantification performance of miniaturized Raman system. **a), b)** Image on CMOS sensor of the SERS spectrum of BPE deposited on nano pillars based SERS substrate at a concentration of 100µM; signal measured laser spot size on the sample of 10µm (a) and 100µm (b), **c)** SERS spectrum of BPE obtained after the averaging of 3 rows on CMOS image in Figure 1a, **d)** SERS spectrum of BPE obtained after the averaging of 20 rows on CMOS image in Figure 1b, **e)** illustration of dynamic QE variation of pixels on CMOS sensor, **f), g)** fluorescence spectrum from glass cover slide excited by laser with excitation wavelength 785nm obtained after averaging of 10 repetitions without pixel averaging method (f) and with pixel averaging method (g) demonstrating improved SNR ratio, **h)** SORS like sample illumination layout with off-axis laser beam delivery and on-axis Raman beam collection that serves dual purpose: avoiding back reflections to diode laser and minimization of fluorescence impact on the out of focus signals, **i)** photograph of miniaturized Raman spectrometer measuring whiskey content through the glass bottle, **j)** Raman spectrum of whiskey measured by miniaturized Raman system through the glass bottle with on-axis (black curve) and off-axis (red curve) laser beam delivery that demonstrates SORS benefit to avoid the contribution of fluorescence from glass bottle (upper right image is a screen shot from CMOS sensor in the case of off-axis laser beam delivery), **k), n)** Raman spectra of water-ethanol solution (40% of ethanol) with different concentrations of methanol (variation between 0-40%) in the range 400-2300cm⁻¹ (k) and 2750-4000cm⁻¹ (n), **i), m)** result of PLS calibration for methanol quantification based on the Raman data in the range 400-2300cm⁻¹, **o), p)** result of PLS calibration for methanol quantification based on the Raman data in the range 2750-4000cm⁻¹.

Application II: Quantification of nutrients and metabolites during fermentation. In industrial production of chemicals by microbial fermentation, the volumes of bioreactors and production costs are often very large, and the reproducibility and the quality of the products are crucial features that must be always ensured. In the pharmaceutical field, for instance, regulatory agencies have been encouraging manufacturers to innovate in the field of process control and monitoring to ensure a sufficient product quality, as demonstrated by the introduction of the Process Analytical Technology (PAT) framework by the U.S. Food and Drug Administration in 2004⁸². The PAT framework outlined a strategy for a continuous control of manufacturing processes through the monitoring of critical process parameters and critical quality attributes to ensure that the product is safe and effective. The concept of continuous process monitoring is also part of the Quality by Design (QbD) paradigm, which approaches the process validation through a continuous improvement of the manufacturing process⁸³. Since a continuous control of the fermentation process is demonstrated to lead to a better product quality, monitoring and controlling tools are being continuously explored and improved. These tools include probes for temperature, pH, dissolved gases and organic carbon, and spectroscopic probes, also including Raman probes^{84,85}.

Besides commercial devices recently developed for monitoring of nutrients and metabolites through Raman^{86, 87}, several examples of on-line and off-line Raman process monitoring of industrial fermentation of bacteria, fungi and mammalian cells⁸⁸ have been reported in literature over the last years. Nutrients, metabolites and by-products have been monitored to gain better knowledge of the fermentation process and to control the feeding strategy⁸⁹⁻⁹¹. The yield of metabolites of interest has been predicted with the aid of complex modeling

systems, as in the case of glycoproteins produced by Chinese hamster ovary⁹², or, in fewer cases, with direct detection of a strongly Raman active compound⁹³.

We used miniaturized Raman spectrometer to perform off-line quantification of pHCA produced during *E. coli* culture, directly measuring the Raman signal of liquid samples of bacterial supernatant (**Figure 3g**). Calibration samples were collected for pHCA, glucose, MgSO₄ and Na₂HPO₄, (**Figure 3a, Figure S22, Table S3**) which were found to be the main contributions to the supernatant spectra at the beginning (black curve) and the end (red curve) of fermentation process (**Figure 3b**). By applying the PLS calibration model to real supernatant samples, the concentration of pHCA was found to increase, whereas glucose decreased over time, as also confirmed by the close correlation with HPLC results (**Figure 3d, 3e**). Also, MgSO₄ and Na₂HPO₄ were consumed over time (**Figure 3f**), although no additional technique was used to validate Raman quantification in this case. Presented results demonstrate extremely high sensitivity of miniaturized Raman spectrometer with LoD for pHCA around 0.01g/L and LoD for glucose around 1g/L. To the best of our knowledge, the lowest previously reported LoD for glucose that was obtained by research grade Raman spectrometer with deep cooling CCD was around 0.55-8g/L^{94,95}. It is important to mention that due to real-time laser intensity calibration and pixel-to-pixel QE compensation implemented in the miniaturized Raman spectrometer we managed to obtain pHCA and glucose quantification error comparable with HPLC (**Figure 3d, 3e**). The presented acquisition method could be applied to more complex cases, such as the quantification of products and nutrients in tank fermentation, and, together with SERDS strategy for fluorescence reduction, could represent a high performance, compact and affordable solution for real time, at-line monitoring of bacterial fermentation. As an example, multiple miniaturized Raman spectrometers could be installed at different tank locations (top, middle and bottom) to monitor fermentation processes more efficiently. **Figure 3h, 3i, S23, Table S4** demonstrate quantification of Serine produced during *E. coli* culture, directly measuring the Raman signal of liquid samples of bacterial supernatant. Here, Raman spectra contained significant fluorescence background (**Figure 3c**), however, we still were able to get quantification error comparable with HPLC⁹⁶ (**Figure 3h,3i**).

Application III: *in-vivo* skin measurements. *In-vivo* skin measurements are typically associated with complex Raman instrumentation that requires a deep cooling sensor due to the low Raman cross section of skin, especially at a depth of more than 100 μ m^{97,98}. Additionally, *in-vivo* skin measurements require the development of an immersion probe with high NA that can provide a small laser spot size in depth of tissue. A small laser spot size will improve the ratio between Raman and fluorescence signal. This happens due to a non-linear saturation of the fluorescence signal and a linear growth of the Raman signal when the laser power is increased⁹⁹. Moreover, it is preferable to produce the last optical element of the probe from fused silica. This will generate optimized conditions for laser/Raman beam propagation in/out of the stratum corneum skin layer and provide a matching of the reflection index between the last optical surface of the probe and the skin media⁹⁷. An extremely compact version of a skin probe is shown in **Figure 3j-3k**. Our probes can be optimized for skin measurements at different depths; between 0 to 150 μ m. When our miniaturized Raman spectrometer is equipped with this probe, we were able to collect Raman spectra of skin at a depth of 10-20 μ m with SNR better than 500:1 (1sec exposure time, 5 repetitions). Typical application examples on studying anti-sun cream penetration and water content are shown in **Figure 3n,3p,3r** respectively. **Figures 3n-s** demonstrate Raman spectra obtained from two lasers (785 and 675nm) at different skin areas (finger, hand and cheek). Water content difference at different skin areas could be clearly seen by intensity ratio of CH (2800-3000cm⁻¹) and OH (3100-3500cm⁻¹) bands. Compared to similar previous experiments^{97,98} our *in-vivo* Raman measurements of skin seem to have unprecedented high SNR (500:1, 1 sec exposure time, 5 repetitions). We believe that the presented instrumentation could be applied for numerous *in-vivo* applications in the future including (i) skin disease diagnostics, (ii) skin aging, (iii) determination of molecular concentration profiles from the skin surface into the dermis, (iv) measuring of the distribution of intrinsic skin constituents (amino acids, sweat constituents, lipids, proteins, water), (v) skin penetration and permeation of topical formulations, (vi) distinguishing of the difference between volar forearm skin, cheek, forehead, scalp, axilla, and other.

Figure 3. Miniaturized Raman spectrometer applied for fermentation monitoring and *in-vivo* skin measurements. **a)** Raman spectra of the main components contributing to the Raman signal of bacterial supernatant, **b)** Raman signal of bacterial supernatant at 0 and 26.5 hours of culture, **c)** Raman spectra of bacterial supernatant containing Serine at 0 to 10 hours of culture collected with step of 1 hour; pink band indicates spectral region with prominent Serine peaks, **d), e), f)** Raman and HPLC quantification of (d) pHCA, (e) glucose and (f) MgSO₄/Na₂HPO₄ in bacterial supernatant (each Raman point is the average of triplicate acquisitions, whereas each HPLC point is the result of duplicate injections), **g)** photograph of miniaturized Raman spectrometer during measurements of fermentation samples, **h), i)** Raman and HPLC quantification of Serine in bacterial supernatant (each Raman and HPLC point is the average of triplicate acquisitions and injections, respectively), **j)** optical schema of the Raman probe with high NA (0.95) developed for *in-vivo* skin measurements, **k)** photograph of “skin probe”, **l), m)** CMOS image of the measurement process of skin *in-vivo*, demonstrating sharp focusing of the Raman signal in the vertical dimension of the sensor under laser excitation wavelength of 785nm (l) and 675nm (m), **n), o), p), q), r), s)** Raman spectra of normal skin collected at the depth of 10–20μm under a laser excitation wavelength of 785nm (n,p,r) and 675nm (o,q,s); pink bands indicate spectral regions of CH and OH peaks; Raman spectra were collected on finger (n,o), hand (p,q) and cheek (r,s); spectrum color represents different probe locations over the skin area around 1cm².

The optical design of our miniaturized Raman spectrometer allows confocal measurements because it utilizes a cross slit confocality concept⁹⁹ as shown in **Figure 4a, 4b**. This feature helps to separate out of focus layers, which is beneficial for typical handheld Raman applications where the contribution from sample packaging or glass needs to be minimized. However, cross slit design also allows us to target confocal Raman microscopy applications when the device is additionally equipped with a three-dimensional motorized stage and a white light microscopy module (**Figure 4c**). The lateral resolution of our miniaturized Raman microscope was tested on polystyrene beads with a 1μm diameter (**Figure 4d**). Axial resolution was tested on the surface of a SERS substrate with BPE analyte at a concentration 100μM (**Figure 4e**). Cross sections in lateral and axial dimensions are represented in **Figure 4f**. They demonstrate a lateral resolution of around 1μm and an axial resolution of around 2μm, indicating a diffraction limited performance in both dimensions. To the best of our knowledge, the presented miniaturized Raman microscope is the smallest reported confocal Raman system that has been designed without compromising on basic performance. Below we present two challenging Raman microscopy applications that typically require research grade systems with deep cooling sensors.

Application IV: quantification of anti-cancer drug via SERS mapping. Therapeutic drug monitoring (TDM) can improve clinical care when using drugs with pharmacokinetic variability and a narrow therapeutic window. Rapid, reliable, and easy-to-use detection methods are required to decrease the time of analysis and can also

enable TDM in resource-limited settings or even at the bedside. Monitoring methotrexate (MTX), an anticancer drug, is critical since it is needed to follow the drug clearance rate and decide how to administer the rescue drug, leucovorin (LV), to avoid toxicity and even death. It has been shown that nanopillar-assisted separation (NPAS) method using SERS mapping by research grade Raman microscope with deep cooling EMCCD allows to measure MTX in PBS in the linear range of 5–150 μM with $\text{LoD} = 5\mu\text{M}$, $\text{LoQ} = 25\mu\text{M}$ ¹⁰⁰. Here, we also used NPAS method with SERS mapping of the SERS chip surface according to the methodology described in the original publication¹⁰⁰. Typical SERS maps of SERS substrates measured by our miniaturized Raman microscope are shown in **Figure 4g**; total measurement time per chip was around 15 mins with exposure time of 0.1 sec per spectrum. In total, 24 SERS chips were used in this study following the NPAS procedure (**Figure 4h, 4i**). Calibration samples of MTX diluted in PBS were prepared in the range 0-75 μM . SERS spectra of MTX obtained after the averaging of SERS signals collected by mapping of the chip surface are shown in **Figure 4j**. Result of PLS calibration for MTX quantification is shown in **Figure 4k, 4l** demonstrating improved $\text{LoD} = 3\mu\text{M}$, $\text{LoQ} = 20\mu\text{M}$ in comparison to previously reported data¹⁰⁰ (see **Figure S24-S29, Table S5**).

Application V: *in-vitro* bacteria identification. The worldwide increase of antimicrobial resistance (AMR) is a serious threat to human health. To avert the spread of AMR, fast reliable diagnostics tools that facilitate optimal antibiotic stewardship are an unmet need. In this regard, Raman spectroscopy promises rapid label- and culture-free identification and antimicrobial susceptibility testing (AST) in a single step. It was recently shown that a Raman-based setup could distinguish bacteria on the species level with more than 96% accuracy when machine learning techniques were combined with a novel data-augmentation algorithm.⁴ In the current study, we used our miniaturized Raman microscope to measure exactly the same bacterial isolates as in the previous publication, where a research grade Raman microscope with deep cooling CCD was used [30] (**Figure 4n**) following identical sample preparation procedures and data analysis (see **Figure S30, S31**). Raman spectra of different bacteria obtained after the averaging of the Raman signal from Raman maps are shown in **Figure 4m**. A total of four Raman maps were collected per single bacteria isolate: three maps contained regions with individual bacteria, and a fourth map was collected as a background (**Figure 4p**). A confusion matrix of bacteria identification built as a result of machine learning data analysis of Raman maps of bacteria is shown in **Figure 4q**. The data obtained with the mini-Raman system show an accuracy of bacteria identification that is comparable to previously reported results from a research-grade Raman setup (overall classification accuracy 98.6%).⁴ However, here we have, thanks to the performance of the spectrometer, been able to reduce the exposure time down to 0.2 sec per spectrum in comparison to 1 sec and 10 repetitions per spectrum in our previous publication while maintaining comparable or higher SNR of raw Raman spectra of bacteria (**Figure S30, S31**).

Figure 4. Miniaturized Raman system applied for biomedical Raman microscopy applications. **a)** image from the CMOS sensor of the SERS signal of BPE; the zoomed region shows that the spectrum is compressed into one row on the sensor, **b)** illustration of the cross slit design of miniaturized Raman spectrometer that is capable for confocal measurements, **c)** optomechanical design of miniaturized Raman microscope based on miniaturized Raman spectrometer, **d)** Raman microscopy image of polystyrene beads at the size of 1µm obtained with Zeiss objective 100x, NA=0.95, **e)** depth scan by our miniaturized Raman microscope (equipped with Zeiss objective 100x, NA=0.95) through the surface of SERS substrate with BPE analyte at concentration 100µM, **f)** axial (black curve) and lateral intensity distribution of Raman signal as a function of sample displacement (dotted white lines in Figure 4d, 4e indicate areas used for plotted axial and lateral intensity profiles); data demonstrate diffraction limited spatial resolution, **g)** SERS maps of MTX deposited on silver coated NP SERS substrates at concentration of 25µM, **h)** photographs demonstrating the process of analyte deposition, **j)** SERS spectra of MTX at different concentrations (0 - 75 µM) obtained after the averaging of SERS signals collected by mapping of the SERS chip, **k), l)** result of PLS calibration for MTX quantification, **m)** Raman spectra of different bacteria obtained after the averaging of the Raman signal from Raman maps, **n)** photograph of miniaturized Raman microscope during Raman mapping of bacteria samples on CaF2 cover glass, **o)** CMOS image of the measurement process of bacteria demonstrating sharp focusing of Raman signal in vertical dimension of the sensor under a laser excitation wavelength of 785nm, **p)** microscopy image of bacteria deposited on CaF2 cover glass with areas selected for Raman mapping of bacteria (areas 1-3) and Raman mapping of cover glass background (area 4), **q)** confusion matrix of bacteria identification built as a result of machine learning data analysis of Raman maps of bacteria.

Discussion

Our basic strategy is centered around a built-in and real-time calibration of Raman shift and laser intensity, facilitated by multiple data processing algorithms. Many of these algorithms rely on information provided by the reference channel. Our proof-of-concept results in **Figure 1a-1q**, **Figure 2a-2j** demonstrate a solution to some of the most pressing problems related to Raman miniaturization. We show that the need for stabilization of laser temperature and power can be circumvented by integration of a reference channel. Additionally, our miniaturization strategy provides: a reduction of sensor dark noise, a compensation on pixel-to-pixel QE variation, a laser optical isolation and maintenance of a high spectral resolution. As an additional feature, our strategy brings forward valuable SERDS and SORS functions. In conclusion, we have been able to miniaturize a Raman spectrometer down to several centimeters (optical part dimensions: 7x2x0.8cm) and have achieved excellent sensitivity (LoD down to 0.07% of methanol in water-ethanol solution), a low power consumption of

around 2 Watts, perfect wavenumber ($\pm 1.5\text{cm}^{-1}$) and intensity calibration ($\pm 1\%$) combined with high spectral resolution of around 7cm^{-1} within the broad spectral range $400\text{-}4000\text{cm}^{-1}$. A comparison of optical and mechanical parameters of the proposed miniaturized Raman spectrometer with typical handheld/benchtop Raman spectrometers and Raman microscopes is presented in Table S6. In comparison with typical handheld Raman units we show improvements in majority of important parameters like spectral resolution, SNR, operation readiness time, wavenumber and intensity calibration accuracy. Moreover, we conclude that overall performance is competitive with high-end benchtop Raman spectrometers and microscopes. The high performance and vast versatility offered by our approach facilitate simple integration into various instruments and applications.

Our proposed concept is currently limited with respect to spectrum registration in low frequency Raman shifts (below 400cm^{-1}) due to laser wavelength drift that does not allow us to apply ultra-steep Raman edge filters close to the laser line. However, precise selection of the diode lasers with known laser wavelength drift versus operation temperature will allow us to apply ultra-steep filters in the next iterations of the system design. Another current limitation is the inability to use long exposures (more than a few seconds) because of spectrum blurring due to laser wavelength drift and high dark noise of the non-cooled sensor. This issue may be partially mitigated by repeated measurements and by applying mode hop deconvolution on smaller exposures with subsequent averaging. This also has the benefit of reducing pixel-to-pixel QE variation effects. However, for extremely weak Raman signals, it seems unpractical to apply very long effective exposure times by having many repetitions, due to dark noise of the sensor. Therefore, certain laser temperature stabilization time may be applied before measurements to be able to reach long exposure times without significant loss of resolution due to spectral mode-hop.

We have exemplified several use cases that all rely on challenging Raman spectroscopy. These include quantification of chemicals at low concentration (through glassware such as bottles or with the use of SERS), *in-vivo* skin measurements and *in-vitro* identification of bacteria (**Figure 2i-2p**, **Figure 3-4**). Application cases IV and V demonstrate that our miniaturized Raman microscope ideally suits for SERS mapping and bacteria mapping applications and provides advantages in key Raman microscopy requirements such as LoD, LoQ, mapping speed and mapping resolution, system size and affordability. Other potential use cases include (i) consumer level chemical characterization of pharmaceuticals, chemicals and food products in order to determine chemical purity and chemical composition, (ii) quality control of chemical products and kinetic monitoring of chemical processes in the industry, (iii) identification of drugs, explosives, toxic liquids and gases by the law enforcement and customs authorities as well as military. Additionally, the proposed technology could be effectively applied on drones, robots and even in space missions due to its small size and weight, low power consumption and autonomous calibration of Raman shift and laser intensity.

We foresee that further development of our technology will allow us to miniaturize the entire Raman module down to a size of $10\times 15\times 3\text{mm}$ with a calculated spectral resolution of around 18cm^{-1} (**Figure S32**, **S33**). If realized this size becomes very attractive for direct integration into smartphones as an in-built chemical analyzer. We believe that our concept can also be applied in different modifications of bulky dispersive Raman spectrometers, SHRS spectrometers and on-chip dispersive and FT spectrometers. This may boost optics miniaturization even further because all the above-mentioned concepts still rely on the need of wavelength and power stabilized lasers. In conclusion we see our miniaturization strategy as a facilitator for both miniaturizing and democratizing Raman spectrometers, making Raman spectroscopy more accessible to researchers as well as consumers.

Methods

Materials, sample preparation and measurements.

Application I: Quantification of toxic methanol in vodka. Methanol, ethanol and water with the purity of 99.9%, 99.8% and 99.9% were used in this research (manufactured by Sigma-Aldrich, CAS Numbers 67-56-1, 64-17-5 and 7732-18-5, respectively). The temperature of the liquid samples was $25\pm 0.4^\circ\text{C}$. The solution of vodka was prepared at concentration ratio 40:60 for ethanol and water. The concentration of components in the methanol-vodka solutions was changed from 0% to 40% of methanol (in volume %). Diluted samples at concentrations 0, 0.1, 0.25, 0.5, 1, 5, 10, 20, 40% with $\pm 0.1\%$ dilution error was prepared with the usage of Eppendorf Pipette Research Plus (volume $100\text{-}1000\mu\text{L}$). Each sample was stored in glass vials with a screw cap (volume 1.5ml). The whiskey bottle used in the studies was produced by Johnnie Walker, Red Label. Whiskey contains 40% of ethanol. The thickness of glass bottle at the measurement area was $\sim 2.7\text{mm}$.

Application II: Quantification of nutrients and metabolites during fermentation. Stock solutions of 100 mM pHCA and 100 mM CA were freshly prepared in EtOH 99%. For Raman calibration standards, pHCA was diluted in control supernatant (obtained from a non-pHCA producing *E. coli* strain (CBJ786)). In addition, for Raman and HPLC experiments the analytes were diluted in M9 medium. HCl 32% was used for acidification of samples and DCM as the organic phase for LLE. Stock solutions of 50 mM Phe and 50 mM Tyr were prepared in water and NaOH at pH ~14 respectively. *E. coli* strains (CBJ800, CBJ786, CBJ951, CBJ789), expressing TAL and PAL-encoding genes from IPTG-inducible promoters, were grown in M9 medium with 1% glucose, 2 mM Tyr and/or 2 mM Phe, 1 mM IPTG and antibiotics for maintenance of plasmids for 22 h as described in our previous work¹⁰¹. For quantification of pHCA produced by *E. coli*, bacterial supernatant samples were obtained from each strain by centrifugation (10 min at 10 000g, 4 °C), and filtration through 0.2 µm filters. The concentration of pHCA in samples was measured with reversed-phase HPLC by separation on a HS-F5 column (Sigma-Aldrich, St Louis, MO, USA) with previously described mobile phases (ammonium formate buffer and acetonitrile), with an overall analysis time of approximately 15 min per sample¹⁰¹. The absorbance was measured at 333 nm for pHCA¹⁰². Aqueous solutions were prepared with ultra-pure water obtained from a Milli-Q purification system (Millipore Corporation, Billerica, MA, USA), and all the chemicals were purchased from Sigma-Aldrich (St Louis, MO, USA). For Raman determination of pHCA in bacterial supernatant, *E. coli* samples (CBJ 800), genetically modified to produce pHCA, were cultured in growth medium according to the methods described in our previous research^{101,102}. Bacterial aliquots were centrifuged and filtered at 0, 1.5, 4.5, 6, 7.5, 9, 11.25, 23 and 26.5 h, with the purpose of monitoring the pHCA production and the nutrient consumption at several time points throughout the culture. The concentrations of pHCA and glucose were also determined through HPLC, with the methods described in our previous publication¹⁰¹.

Raman measurements were carried out at fixed laser power on the sample of 100mW (from laser excitation wavelength 785nm) and 30mW (from laser excitation wavelength 675nm)A custom Raman probe with NA = 0.08 was used in this study which provides an estimated laser spot size of around 10µm on the sample. Each spectrum was averaged over 10 repetitions at an exposure time of 1 s each. A sample volume of 500 µL was poured into a glass vial with screw cap (volume 1.5ml). The Raman signal of liquid samples was collected by focusing the laser beam in the middle of the vial through a bottom window, and each sample was collected in triplicates.

Application III: *in-vivo* skin measurements. Anti-sun cream Sollotion SPF30 produced by DermaPharm A/S was used in this study. Cream was applied on human palm that was previously cleaned with water-soup solution to avoid skin surface contamination by dust. The skin analysis was performed on own skin by the main authors and thus does not require specific ethics permission.

Microscopy test on PS beads. PS beads with size of 1µm in the form of aqueous suspension were purchased on Merck (MDL number: MFCD00243243). Suspension was deposited on polished stainless-steel surface for Raman microscopy mapping.

Application IV: quantification of anti-cancer drug via SERS mapping. MTX (98% purity) was initially dissolved in 50 µL of 1 M NaOH, and 2 mM stock solution was prepared in phosphate-buffered saline (PBS), pH 7.4, which was aliquoted and stored at -20 °C until further use. The MTX stock solutions were used to freshly prepare standards in PBS. The MTX standard solutions in PBS was mixed with methanol (MeOH) in various concentration of MTX: 0, 5, 10, 25, 50, 75 µM. Solvents, chemicals, and samples were of analytical grade and purchased from Sigma-Aldrich (St. Louis, MO, USA). Fabrication of the AgNP SERS substrates and NPAS/SERS detection was described in our previous publication.

Application V: *in-vitro* bacteria identification. The bacteria were obtained from overnight on agar plate cultures which were sealed with parafilm and stored at 5 °C until sample preparation. Storage time varied but was not found to result in spectral changes to strain or phenotype characteristics. All other sample preparation conditions were kept consistent between samples. Test samples were prepared separately from samples used for training, to ensure classification was not influenced by differences in sample preparation. To prepare samples for Raman measurement, a sample was simply transferred from a single colony directly to a sterilized CaF2 Raman-grade objective slide. Detailed description of sample preparation and data analysis can be found in our previous publication.⁴

Data availability

The source data for Figures 1–4 are provided with the paper. The data that support the other findings of this study are available from the corresponding author upon reasonable request.

Code availability

Matlab code for mathematical algorithms is provided with the paper. The code that support the other findings of this study is available from the corresponding author upon reasonable request.

References

- 1 Pezzotti G. Raman spectroscopy in cell biology and microbiology. *J Raman Spectrosc* 2021; **52**: 2348–2443.
- 2 Downes A, Mouras R, Bagnaninchi P, Elfick A. Raman spectroscopy and CARS microscopy of stem cells and their derivatives. *J Raman Spectrosc* 2011; **42**: 1864–1870.
- 3 Auner GW, Koya SK, Huang C, Broadbent B, Trexler M, Auner Z *et al.* Applications of Raman spectroscopy in cancer diagnosis. *Cancer Metastasis Rev* 2018; **37**: 691–717.
- 4 Thomsen BL, Christensen JB, Rodenko O, Usenov I, Grønnemose RB, Andersen TE *et al.* Accurate and fast identification of minimally prepared bacteria phenotypes using Raman spectroscopy assisted by machine learning. *Sci Rep* 2022; **12**: 16436.
- 5 Stöckel S, Kirchhoff J, Neugebauer U, Rösch P, Popp J. The application of Raman spectroscopy for the detection and identification of microorganisms. *J Raman Spectrosc* 2016; **47**: 89–109.
- 6 Wang L, Liu W, Tang J-W, Wang J-J, Liu Q-H, Wen P-B *et al.* Applications of Raman Spectroscopy in Bacterial Infections: Principles, Advantages, and Shortcomings. *Front Microbiol* 2021; **12**. doi:10.3389/fmicb.2021.683580.
- 7 Ho C-S, Jean N, Hogan CA, Blackmon L, Jeffrey SS, Holodniy M *et al.* Rapid identification of pathogenic bacteria using Raman spectroscopy and deep learning. *Nat Commun* 2019; **10**: 4927.
- 8 Doty KC, Muro CK, Bueno J, Halámková L, Lednev IK. What can Raman spectroscopy do for criminalistics? *J Raman Spectrosc* 2016; **47**: 39–50.
- 9 Petersen M, Yu Z, Lu X. Application of Raman Spectroscopic Methods in Food Safety: A Review. *Biosensors* 2021; **11**: 187.
- 10 Downes A, Elfick A. Raman Spectroscopy and Related Techniques in Biomedicine. *Sensors* 2010; **10**: 1871–1889.
- 11 Kong K, Kendall C, Stone N, Notingher I. Raman spectroscopy for medical diagnostics — From in-vitro biofluid assays to in-vivo cancer detection. *Adv Drug Deliv Rev* 2015; **89**: 121–134.
- 12 Lieber CA, Majumder SK, Billheimer D, Ellis DL, Mahadevan-Jansen A. Raman microspectroscopy for skin cancer detection in vitro. *J Biomed Opt* 2008; **13**: 024013.
- 13 Ramírez-Elías MG, González FJ. Raman Spectroscopy for In Vivo Medical Diagnosis. In: *Raman Spectroscopy*. InTech, 2018 doi:10.5772/intechopen.72933.
- 14 Hanlon EB, Manoharan R, Koo T-W, Shafer KE, Motz JT, Fitzmaurice M *et al.* Prospects for in vivo Raman spectroscopy. *Phys Med Biol* 2000; **45**: R1–R59.
- 15 Cordero E. In-vivo Raman spectroscopy: from basics to applications. *J Biomed Opt* 2018; **23**: 1.
- 16 Motz JT, Gandhi SJ, Scepanovic OR, Haka AS, Kramer JR, Dasari RR *et al.* Real-time Raman system for in vivo disease diagnosis. *J Biomed Opt* 2005; **10**: 031113.
- 17 Culka A, Košek F, Drahotka P, Jehlička J. Use of miniaturized Raman spectrometer for detection of sulfates of different hydration states – Significance for Mars studies. *Icarus* 2014; **243**: 440–453.
- 18 Košek F, Culka A, Drahotka P, Jehlička J. Applying portable Raman spectrometers for field discrimination of sulfates: Training for successful extraterrestrial detection. *J Raman Spectrosc* 2017; **48**: 1085–1093.

- 19 Dickensheets DL, Wynn-Williams DD, Edwards HGM, Schoen C, Crowder C, Newton EM. A novel miniature confocal microscope/Raman spectrometer system for biomolecular analysis on future Mars missions after Antarctic trials. *J Raman Spectrosc* 2000; **31**: 633–635.
- 20 Kim S, Joo J-H, Kim W, Bang A, Choi HW, Moon SW *et al.* A facile, portable surface-enhanced Raman spectroscopy sensing platform for on-site chemometrics of toxic chemicals. *Sensors Actuators B Chem* 2021; **343**: 130102.
- 21 Wang W, Ma P, Song D. Applications of surface-enhanced Raman spectroscopy based on portable Raman spectrometers: A review of recent developments. *Luminescence* 2022; **37**: 1822–1835.
- 22 Guo J, Liu Y, Ju H, Lu G. From lab to field: Surface-enhanced Raman scattering-based sensing strategies for on-site analysis. *TrAC Trends Anal Chem* 2022; **146**: 116488.
- 23 Bratchenko IA, Bratchenko LA, Moryatov AA, Khristoforova YA, Artemyev DN, Myakinin OO *et al.* In vivo diagnosis of skin cancer with a portable Raman spectroscopic device. *Exp Dermatol* 2021; **30**: 652–663.
- 24 Kang JW, Park YS, Chang H, Lee W, Singh SP, Choi W *et al.* Direct observation of glucose fingerprint using in vivo Raman spectroscopy. *Sci Adv* 2020; **6**. doi:10.1126/sciadv.aay5206.
- 25 Coffey P, Smith N, Lennox B, Kijne G, Bowen B, Davis-Johnston A *et al.* Robotic arm material characterisation using LIBS and Raman in a nuclear hot cell decommissioning environment. *J Hazard Mater* 2021; **412**: 125193.
- 26 Pinto M, Zorn KC, Tremblay J-P, Desroches J, Dallaire F, Aubertin K *et al.* Integration of a Raman spectroscopy system to a robotic-assisted surgical system for real-time tissue characterization during radical prostatectomy procedures. *J Biomed Opt* 2019; **24**: 1.
- 27 Ashok PC, Giardini ME, Dholakia K, Sibbett W. A Raman spectroscopy bio-sensor for tissue discrimination in surgical robotics. *J Biophotonics* 2014; **7**: 103–109.
- 28 Ilchenko O, Pilhun Y, Kutsyk A. Towards Raman imaging of centimeter scale tissue areas for real-time opto-molecular visualization of tissue boundaries for clinical applications. *Light Sci Appl* 2022; **11**: 143.
- 29 Mu T, Li S, Feng H, Zhang C, Wang B, Ma X *et al.* High-Sensitive Smartphone-Based Raman System Based on Cloud Network Architecture. *IEEE J Sel Top Quantum Electron* 2019; **25**: 1–6.
- 30 Kita DM, Miranda B, Favela D, Bono D, Michon J, Lin H *et al.* High-performance and scalable on-chip digital Fourier transform spectroscopy. *Nat Commun* 2018; **9**: 4405.
- 31 Zhang L, Chen J, Ma C, Li W, Qi Z, Xue N. Research Progress on On-Chip Fourier Transform Spectrometer. *Laser Photon Rev* 2021; **15**: 2100016.
- 32 Xiao Ma, Mingyu Li, Jian-Jun He. CMOS-Compatible Integrated Spectrometer Based on Echelle Diffraction Grating and MSM Photodetector Array. *IEEE Photonics J* 2013; **5**: 6600807–6600807.
- 33 Ryckeboer E, Nie X, Subramanian AZ, Martens D, Bienstman P, Clemmen S *et al.* CMOS-compatible silicon nitride spectrometers for lab-on-a-chip spectral sensing. In: Vivien L, Pavesi L, Pelli S (eds). . 2016, p 98911K.
- 34 Heaton HI. Interferometric Raman spectrometry with fiber waveguides. *Appl Opt* 1997; **36**: 6739.
- 35 Pol Van Dorpe PP. Optical spectrometer with matched étendue. 2018.<https://patents.google.com/patent/US9909992B2/en>.
- 36 Hu T, Zhang X, Zhang M, Yan X. A high-resolution miniaturized ultraviolet spectrometer based on arrayed waveguide grating and microring cascade structures. *Opt Commun* 2021; **482**: 126591.
- 37 Ismail N, Choo-Smith L-P, Wörhoff K, Driessen A, Baclig AC, Caspers PJ *et al.* Raman spectroscopy with an integrated arrayed-waveguide grating. *Opt Lett* 2011; **36**: 4629.
- 38 Harwit M, Sloane NJA. Instrumental Considerations. In: *Hadamard Transform Optics*. Elsevier, 1979, pp

- 109–145.
- 39 Kita DM, Miranda B, Favela D, Bono D, Michon J, Lin H *et al.* High-performance and scalable on-chip digital Fourier transform spectroscopy. *Nat Commun* 2018; **9**: 4405.
- 40 Vunckx K, Geelen B, Garcia Munoz V, Lee W, Chang H, Van Dorpe P *et al.* Towards a miniaturized application-specific Raman spectrometer. In: Kim MS, Cho B-K, Chin BA (eds). *Sensing for Agriculture and Food Quality and Safety XII*. SPIE, 2020, p 8.
- 41 Korinth F, Schmäzlin E, Stiebing C, Urrutia T, Micheva G, Sandin C *et al.* Wide Field Spectral Imaging with Shifted Excitation Raman Difference Spectroscopy Using the Nod and Shuffle Technique. *Sensors* 2020; **20**: 6723.
- 42 Manzoni C, Comelli D, Cerullo GN, Ardini B, Vanna R, Bassi A *et al.* A high-throughput Fourier-transform wide-field hyperspectral microscope for fluorescence and Raman imaging. In: Groves R, Liang H (eds). *Optics for Arts, Architecture, and Archaeology VIII*. SPIE, 2021, p 8.
- 43 Barnett PD, Angel SM. Miniature Spatial Heterodyne Raman Spectrometer with a Cell Phone Camera Detector. *Appl Spectrosc* 2017; **71**: 988–995.
- 44 Waldron A, Allen A, Colón A, Carter JC, Angel SM. A Monolithic Spatial Heterodyne Raman Spectrometer: Initial Tests. *Appl Spectrosc* 2021; **75**: 57–69.
- 45 Feng Z, Xia G, Lu R, Cai X, Cui H, Hu M. High-Performance Ultra-Thin Spectrometer Optical Design Based on Coddington's Equations. *Sensors* 2021; **21**: 323.
- 46 Xia C, Zeng C, Feng Z. Design of optical system of crossed astigmatism Czerny-Turner spectrometer. In: Zhao H, Liu J, Xu L, Wang Y (eds). *AOPC 2021: Optical Spectroscopy and Imaging*. SPIE, 2021, p 35.
- 47 Dynamic, robust and versatile sensors for diverse applications. *Nat Photonics* 2008; **2**: 157–157.
- 48 Auz B, Bonvallet J, Olmstead T, Rodriguez J. Miniature Raman spectrometer development. In: Mahadevan-Jansen A, Petrich W (eds). *Biomedical Vibrational Spectroscopy 2018: Advances in Research and Industry*. SPIE, 2018, p 36.
- 49 Belay GY, Hoving W, van der Put A, Vervaeke M, Van Erps J, Thienpont H *et al.* Miniaturized broadband spectrometer based on a three-segment diffraction grating for spectral tissue sensing. *Opt Lasers Eng* 2020; **134**: 106157.
- 50 Thomas Rasmussen, Poul Hansen, Bjarke Rose, Ole Jespersen, Nicolai Rasmussen MR. How to Design a Miniature Raman Spectrometer. *Spectroscopy* 2015; **30**.<https://www.spectroscopyonline.com/view/how-design-miniature-raman-spectrometer>.
- 51 Denson SC, Pommier CJS, Denton MB. The Impact of Array Detectors on Raman Spectroscopy. *J Chem Educ* 2007; **84**: 67.
- 52 Cooper JB, Aust J, Stellman C, Chike K, Myrick ML, Schwartz R *et al.* Raman spectroscopy with a low-cost imaging CCD array. *Spectrochim Acta Part A Mol Spectrosc* 1994; **50**: 567–575.
- 53 Smith RM, Rahmer G. Pixel area variation in CCDs and implications for precision photometry. In: Dorn DA, Holland AD (eds). . 2008, p 70212A.
- 54 Kotov I V., Kotov AI, Frank J, Kubanek P, Prouza M, O'Connor P *et al.* Study of pixel area variations in fully depleted thick CCD. In: Holland AD, Dorn DA (eds). . 2010, p 774206.
- 55 Hennelly B, Barton S. Signal to noise ratio of Raman spectra of biological samples. In: Popp J, Tuchin V V., Pavone FS (eds). *Biophotonics: Photonic Solutions for Better Health Care VI*. SPIE, 2018, p 160.
- 56 Sommer L. *Analytical absorption spectrophotometry in the visible and ultraviolet: the principles*. Elsevier, 2012.
- 57 Kobayashi M, Ota T. Spectrometry device and spectrometry method. 2010; : US9435741B2.
- 58 Wang W, Major A, Paliwal J. Grating-Stabilized External Cavity Diode Lasers for Raman Spectroscopy—

- A Review. *Appl Spectrosc Rev* 2012; **47**: 116–143.
- 59 Cooney TF, Skinner HT, Angel SM. Evaluation of External-Cavity Diode Lasers for Raman Spectroscopy. *Appl Spectrosc* 1995; **49**: 1846–1851.
- 60 Saliba SD, Junker M, Turner LD, Scholten RE. Mode stability of external cavity diode lasers. *Appl Opt* 2009; **48**: 6692.
- 61 Angel SM, Carrabba M, Cooney TF. The utilization of diode lasers for Raman spectroscopy. *Spectrochim Acta Part A Mol Biomol Spectrosc* 1995; **51**: 1779–1799.
- 62 John B. Cooper, Philip E. Flecher, Sacharia Albin, Thomas M. Vess and WTW. Elimination of Mode Hopping and Frequency Hysteresis in Diode Laser Raman Spectroscopy: The Advantages of a Distributed Bragg Reflector Diode Laser for Raman Excitation. *Appl Spectrosc* 1995; **49**: 1692–1698.
- 63 Leonhäuser B, Kissel H, Tomm JW, Hempel M, Unger A, Biesenbach J. High-power diode lasers under external optical feedback. In: Zediker MS (ed). . 2015, p 93480M.
- 64 Maiwald M, Erbert G, Klehr A, Sumpf B, Wenzel H, Laurent T *et al*. Reliable operation of 785 nm DFB diode lasers for rapid Raman spectroscopy. In: Zediker MS (ed). . 2007, p 64560W.
- 65 Petermann K. External optical feedback phenomena in semiconductor lasers. *IEEE J Sel Top Quantum Electron* 1995; **1**: 480–489.
- 66 Engelbrecht R, Lins B, Zinn P, Buchtal R, Schmauss B. Line shapes of near-infrared DFB and VCSEL diode lasers under the influence of system back reflections. *Appl Phys B* 2012; **109**: 441–452.
- 67 Goldberg L, Taylor HF, Dandridge A, Weller JF, Miles RO. Spectral Characteristics of Semiconductor Lasers with Optical Feedback. *IEEE Trans Microw Theory Tech* 1982; **30**: 401–410.
- 68 Leisher PO, Li C, Jha AK, Pipe KP, Helmrich JD, Thiagarajan P *et al*. Feedback-Induced Failure of High-Power Diode Lasers. *IEEE J Quantum Electron* 2018; **54**: 1–13.
- 69 Kuwahara H, Onoda Y, Sasaki M, Shirasaki M. An optical isolator for semiconductor lasers in the 0.8 μm range. *Opt Commun* 1981; **40**: 99–104.
- 70 Nguyen HT, Shore BW, Bryan SJ, Britten JA, Boyd RD, Perry MD. High-efficiency fused-silica transmission gratings. *Opt Lett* 1997; **22**: 142.
- 71 Gove RJ. CMOS image sensor technology advances for mobile devices. In: *High Performance Silicon Imaging*. Elsevier, 2020, pp 185–240.
- 72 Aarnoutse PJ, Westerhuis JA. Quantitative Raman Reaction Monitoring Using the Solvent as Internal Standard. *Anal Chem* 2005; **77**: 1228–1236.
- 73 Chalmond B. PSF estimation for image deblurring. *CVGIP Graph Model Image Process* 1991; **53**: 364–372.
- 74 Liu MS, Bursill LA, Prawer S, Beserman R. Temperature dependence of the first-order Raman phonon line of diamond. *Phys Rev B* 2000; **61**: 3391–3395.
- 75 Gebrekidan MT, Knipfer C, Stelzle F, Popp J, Will S, Braeuer A. A shifted-excitation Raman difference spectroscopy (SERDS) evaluation strategy for the efficient isolation of Raman spectra from extreme fluorescence interference. *J Raman Spectrosc* 2016; **47**: 198–209.
- 76 Guo S, Chernavskaia O, Popp J, Bocklitz T. Spectral reconstruction for shifted-excitation Raman difference spectroscopy (SERDS). *Talanta* 2018; **186**: 372–380.
- 77 Dieing T, Hollricher O. High-resolution, high-speed confocal Raman imaging. *Vib Spectrosc* 2008; **48**: 22–27.
- 78 Tateda M, Matsushita H. Optical isolator independent of input polarization direction utilizing a quarter-wave plate. In: *Digest of the LEOS Summer Topical Meetings, 2005*. IEEE, pp 47–48.

- 79 Mosca S, Conti C, Stone N, Matousek P. Spatially offset Raman spectroscopy. *Nat Rev Methods Prim* 2021; **1**: 21.
- 80 Ellis DI, Muhamadali H, Xu Y, Eccles R, Goodall I, Goodacre R. Rapid through-container detection of fake spirits and methanol quantification with handheld Raman spectroscopy. *Analyst* 2019; **144**: 324–330.
- 81 Paine AJ, Dayan AD. Defining a tolerable concentration of methanol in alcoholic drinks. *Hum Exp Toxicol* 2001; **20**: 563–568.
- 82 FDA. Guidance for Industry PAT: A Framework for Innovative Pharmaceutical Development, Manufacturing, and Quality Assurance. *FDA Off Doc* 2004; : 16.
- 83 ICH. Pharmaceutical Development Q8. *ICH Harmon Tripart Guidel* 2009; **8**: 1–28.
- 84 O'Mara P, Farrell A, Bones J, Twomey K. Staying alive! Sensors used for monitoring cell health in bioreactors. *Talanta* 2018; **176**: 130–139.
- 85 Biechele P, Busse C, Solle D, Scheper T, Reardon K. Sensor systems for bioprocess monitoring. *Eng Life Sci* 2015; **15**: 469–488.
- 86 PROCELLICS™ In-Line and Real-Time Bioprocess Raman Analyzer. .
- 87 SUNIL-INA INSTRUMENT LTD. ® Raman RXN3 Analyzer. .
- 88 Esmonde-White KA, Cuellar M, Uerpmann C, Lenain B, Lewis IR. Raman spectroscopy as a process analytical technology for pharmaceutical manufacturing and bioprocessing. *Anal Bioanal Chem* 2017; **409**: 637–649.
- 89 Shope TB, Vickers TJ, Mann CK. The direct analysis of fermentation products by Raman spectroscopy. *Appl Spectrosc* 1987; **41**: 908–912.
- 90 Sivakesava S, Irudayaraj J, Demirci A. Monitoring a bioprocess for ethanol production using FT-MIR and FT-Raman spectroscopy. *J Ind Microbiol Biotechnol* 2001; **26**: 185–190.
- 91 Xu Y, Ford JF, Mann CK, Vickers TJ, Brackett JM, Cousineau KL *et al.* Raman measurement of glucose in bioreactor materials. In: Vo-Dinh T, Lieberman RA, Vurek GG, Katzir A (eds). *Biomedical Sensing, Imaging, And Tracking Technologies II*. 1997, pp 10–19.
- 92 Li B, Ray BH, Leister KJ, Ryder AG. Performance monitoring of a mammalian cell based bioprocess using Raman spectroscopy. *Anal Chim Acta* 2013; **796**: 84–91.
- 93 Cannizzaro C, Rhiel M, Marison I, von Stockar U. On-line monitoring of *Phaffia rhodozyma* fed-batch process with in situ dispersive raman spectroscopy. *Biotechnol Bioeng* 2003; **83**: 668–680.
- 94 Shih C-J, Smith EA. Determination of glucose and ethanol after enzymatic hydrolysis and fermentation of biomass using Raman spectroscopy. *Anal Chim Acta* 2009; **653**: 200–206.
- 95 Ávila TC, Poppi RJ, Lunardi I, Tizei PAG, Pereira GAG. Raman spectroscopy and chemometrics for on-line control of glucose fermentation by *Saccharomyces cerevisiae*. *Biotechnol Prog* 2012; **28**: 1598–1604.
- 96 Jendresen CB, Stahlhut SG, Li M, Gaspar P, Siedler S, Förster J *et al.* Highly Active and Specific Tyrosine Ammonia-Lyases from Diverse Origins Enable Enhanced Production of Aromatic Compounds in Bacteria and *Saccharomyces cerevisiae*. *Appl Environ Microbiol* 2015; **81**: 4458–4476.
- 97 Caspers PJ, Bruining HA, Puppels GJ, Lucassen GW, Carter EA. In Vivo Confocal Raman Microspectroscopy of the Skin: Noninvasive Determination of Molecular Concentration Profiles. *J Invest Dermatol* 2001; **116**: 434–442.
- 98 Nakagawa N, Matsumoto M, Sakai S. In vivo measurement of the water content in the dermis by confocal Raman spectroscopy. *Ski Res Technol* 2010; **16**: 137–141.
- 99 Slipets R, Ilchenko O, Mazzoni C, Tentor F, Nielsen LH, Boisen A. Volumetric Raman chemical imaging of

- drug delivery systems. *J Raman Spectrosc* 2020; **51**: 1153–1159.
- 100 Göksel Y, Zor K, Rindzevicius T, Thorhauge Als-Nielsen BE, Schmiegelow K, Boisen A. Quantification of Methotrexate in Human Serum Using Surface-Enhanced Raman Scattering—Toward Therapeutic Drug Monitoring. *ACS Sensors* 2021; **6**: 2664–2673.
- 101 Morelli L, Seriola L, Centorbi FA, Jendresen CB, Matteucci M, Ilchenko O *et al.* Injection molded lab-on-a-disc platform for screening of genetically modified *E. coli* using liquid–liquid extraction and surface enhanced Raman scattering. *Lab Chip* 2018; **18**: 869–877.
- 102 Morelli L, Zór K, Jendresen CB, Rindzevicius T, Schmidt MS, Nielsen AT *et al.* Surface Enhanced Raman Scattering for Quantification of p -Coumaric Acid Produced by *Escherichia coli*. *Anal Chem* 2017; **89**: 3981–3987.

REVIEWERS' COMMENTS

Reviewer #2 (Remarks to the Author):

The authors have considered the comments and improved the manuscript accordingly. The added Table S6 in the supplementary information really helps to see the properties of the suggested miniaturized Raman spectrometer in comparison to established benchtop or handheld spectrometers. Together with the extensive characterisation of the fabricated prototype the authors clearly make a convincing case for their miniaturized solution. The prototype combines a lot of strategies taken from different areas of optics and electronics to reduce noise, cost and size. In this way the paper describes an interesting case of product development. In my opinion the described prototype will act as a realizable benchmark for the functionality of miniaturized Raman spectrometers and other technological concepts under development will have to measure up to this.

In conclusion I recommend the paper in its current improved version for publication in Nature Communications.